Self-reported history of head injury is associated with cognitive impulsivity on a delay discounting task

Allen M. Todd michael.allen@unco.edu 1
Interian Alejandro 2 3
Reddy Vibha 4
Rodriguez Kailyn 4 5
Myers Catherine E. 5 6
1 School of Psychological Sciences, University of Northern Colorado , Greeley , CO , United States of America
2 Mental Health and Behavioral Sciences, VA New Jersey Health Care System , Lyons , NJ , United States of America
3 Department of Psychiatry, Robert Wood Johnson Medical School, Rutgers University , Piscataway , NJ , United States of America
4 Department of Psychology, Rutgers University School of Graduate Studies , Piscataway , NJ , United States of America
5 Research and Development Service, VA New Jersey Health Care System , East Orange , NJ , United States of America
6 Department of Pharmacology, Physiology & Neuroscience, Rutgers University-New Jersey Medical School , Newark , NJ , United States of America
van den Hoek Daniel
Electronic publication date: 2025 Mar 7
Publication date: 2025
Volume: 13
Electronic Location ID: e19057
Received 2024 Sep 23; Accepted 2025 Feb 5
Copyright: ©2025 Allen et al.
Copyright year: 2025
Copyright holder: Allen et al.
License: This is an open access article distributed under the terms of the Creative Commons Attribution License, which permits unrestricted use, distribution, reproduction and adaptation in any medium and for any purpose provided that it is properly attributed. For attribution, the original author(s), title, publication source (PeerJ) and either DOI or URL of the article must be cited.
License URL: https://creativecommons.org/licenses/by/4.0/

Keywords: Head injury, Impusivity, Undergraduates, Delay discounting, Go no-go task

Funding: Merit Review Award #01 CX001826 from the U. S. Department of Veterans Affairs Clinical Sciences Research and Development Service This work was supported by Merit Review Award #01 CX001826 from the U. S. Department of Veterans Affairs Clinical Sciences Research and Development Service. The funders had no role in study design, data collection and analysis, decision to publish, or preparation of the manuscript.

==============================
Background

Head injuries are a major health care concern that can produce many long lasting cognitive, mental, and physical problems. An emerging literature indicates increased impulsivity in patients with a history of traumatic brain injury (TBI). In a recent study, Veterans with clinically-assessed history of mild TBI had increased cognitive, but not motor, impulsivity. Cognitive impulsivity refers to a preference for smaller immediate rewards (i.e., less willing to wait for larger rewards) while motor impulsivity refers to difficulty inhibiting a motor response. This study extended this work to investigating cognitive and motor impulsivity in a non-clinical sample of putatively healthy undergraduates self-reporting a history of head injury.

Methods

One hundred and sixteen undergraduates, fifty reporting a history of head injury (HI+) and sixty-six reporting no head injury (HI-), participated in an online study via Qualtrics. They completed a series of demographic questionnaires, the UPPS Impulsive Behavior Scale, a computer-based Go/No-go task to assess motor impulsivity, and a computer-based version of the Monetary Choice Questionnaire (MCQ) to assess cognitive impulsivity.

Results

HI+ individuals exhibited cognitive impulsivity, measured as a reduced willingness to wait for a larger delayed reward in the MCQ, as compared to HI- individuals. There were no significant differences in performance on the Go/No-go task between the HI+ and HI- groups. Overall, these findings that a self-reported history of head injury in a non-clinical sample are related to cognitive impulsivity, but not motor impulsivity, are consistent with findings from Veterans with clinically-assessed mild TBI. Future work should assess more details on head injuries to further explore how a head injury relates to cognitive impulsivity.

Introduction

Head injury (HI) resulting in traumatic brain injury (TBI) or concussion is a major health concern especially in youth, adolescents and young adults, and the elderly (Ryan et al., 1996). TBI is a leading cause of death and disability worldwide (Maas et al., 2017; Quaglio et al., 2017), and it is estimated that 50–60 million individuals are affected by TBI annually, and that close to half of the world’s population will sustain a TBI in their lifetime (Maas et al., 2017). In the US, 1.4 million individuals receive emergency room or some other form of medical care for mild brain injuries yearly (Bazarian et al., 2005); however, these reports only include those that seek medical care, so the actual number of head injuries is underestimated.

Ryan et al. (1996) suggested that head injuries peak at three timepoints across the lifespan, one in early childhood (ages one to five), one in adolescence and early adulthood (ages 15–24), and one in old age (over 65 years of age). Researchers have suggested that 23% of college undergraduates (Ryan et al., 1996), or 24% of male and 16% of female college students (Crovitz, Horn & Daniel, 1983), have self-reported a history of head injury. Accordingly, there has been special interest in the effects of head injury in college age students.

Indeed, Kennedy, Krause & O’Brien (2014) reported many issues in college students who had suffered a HI including problems with learning/studying, organization and time management, social issues, and nervousness and anxiety. Furthermore, Hux, Brown & Lambert (2017) grouped consequences of HI into three categories for college students: cognitive, physiological/somatic, and social emotional/affective. Cognitive consequences include deficits in attention, concentration, and organization affecting information processing and may disrupt new learning and memories.

Increased impulsivity is another known consequence in patients with a history of TBI (Goldstein, 1952; Carroll et al., 2014). Oas (1985) defined impulsive behavior as occurring quickly and without forethought and being socially inappropriate or maladaptive, while McAllister (2008) defined impulsivity as the tendency to express spontaneous and excessive behaviors. In general, impulsivity is associated with extraversion, sensation and novelty-seeking, poor planning, disorganization, and a lack of control (Depue & Collins, 1999; Dickman, 1990; Eysenck & Eysenck, 1977; Logan, Schachar & Tannock, 1997; McCrae & Costa Jr, 1990). Several studies suggest that survivors of TBI display increased impulsivity traits (Stuss, 2011; Rodriguez-Bailon, Trivino & Lupianez, 2012; Rochat et al., 2010); for example, Mosti & Coccarro (2018) reported that adults with history of TBI had higher self-reported impulsivity as measured by the Barratt Impulsiveness Scale (BIS; Patton, Stanford & Barratt, 1995) and Life History of Impulsive Behavior (LHIB; Coccaro & Schmidt-Kaplan, 2012).

However, impulsivity is heterogenous by its very nature and includes a range of cognitive and behavioral indicators (Caswell et al., 2015). For example, cognitive impulsivity is defined as a preference for smaller more immediate rewards over larger, more delayed rewards (Ainslie, 1975; Herrnstein, 1981; Logue, 1988; Rachlin, 1989; Rachlin & Green, 1972). Cognitive impulsivity is often assessed through the Monetary Choice Questionnaire (MCQ; Kirby, Petry & Bickel, 1999). Several studies have reported that adults with a history of TBI display less willingness to wait for delayed rewards, compared to participants with no history of TBI (McHugh & Wood, 2008; Richards et al., 1999).

Conversely, motor impulsivity refers to difficulty inhibiting or stopping a prepotent motor response. Motor impulsivity is often assessed through a Go/No-go (GNG) task, in which individuals are challenged to respond as quickly as possible to on-screen targets, while withholding (inhibiting) responses to infrequent foil stimuli. Several studies have reported that history of TBI is associated with poorer performance on GNG (e.g., Gagnon et al., 2006; Dimoska-Di Marco et al., 2011).

However, measures of cognitive and motor impulsivity are not related directly to one another. For example, in a study of non-clinical undergraduates. Caswell et al. (2015) found that motor impulsivity (on a Go/No-go task) was only weakly associated with cognitive impulsivity (on the MCQ). In addition, self-reported impulsivity, as assessed by the Barratt Impulsivity Scale (BIS-11; Stanford et al., 2009), is related to Go/No-go performance (Aichert et al., 2012; Reynolds et al., 2006; Weidacker et al., 2017) but not cognitive impulsivity (Lane et al., 2003; Lansbergen, Schutter & Kenemans, 2007; Reynolds et al., 2006). Another self-report scale, the UPPS Impulsive Behavior Scale (Whiteside & Lynam, 2001) assesses five dimensions of impulsivity—negative urgency, (lack of) premeditation, (lack of) perseverance, sensation seeking, and positive urgency; in one study, UPPS subscores were positively correlated with each other but negatively correlated with cognitive impulsivity assessed via the MCQ (Mulhauser et al., 2019). Other studies have found that negative urgency is associated with both cognitive impulsivity (MCQ) and motor impulsivity (GNG) in healthy adults (e.g., Levitt et al., 2021).

Turning to TBI, results are mixed. Some studies have reported that both cognitive and motor impulsivity are impaired following brain injury (Dimoska-Di Marco et al., 2011; Gunnarsson, Whiting & Sims, 2018). However, in one recent study (Interian et al., 2024) with a well-characterized clinical sample of Veterans, with and without history of mild TBI documented by a careful clinical review, those with TBI history showed decreased willingness-to-wait for the larger delayed reward on MCQ; however, the same participants showed no effect of TBI history on motor impulsivity using GNG. The current study investigated whether a similar dissociation between cognitive impulsivity (on MCQ) and motor impulsivity (on GNG) would be observed in putatively healthy, high-functioning young adults self-reporting history of head injury.

Aim and hypotheses

The aim of the current study was therefore to extend and replicate the prior findings of Interian et al. (2024), in a sample of (putatively healthy, high-functioning) college undergraduates, self-reporting a history of HI. Our hypotheses were that those with self-reported HI (HI+) would show increased cognitive impulsivity, but not motor impulsivity, compared to those with no history of HI (HI-). Such a finding would partially replicate and extend the prior results observed in the clinically-assessed sample from Interian et al. (2024). In addition, we administered a self-report questionnaire, the UPPS, to query impulsive personality traits. Our exploratory hypothesis was that those with self-reported HI would show more negative urgency. We also expected to replicate prior findings (e.g., Levitt et al., 2021) that cognitive impulsivity (on MCQ) would correlate more strongly with negative urgency than with other UPPS sub-scores.

Materials & Methods

Participants

One hundred and sixteen college undergraduates (69% female, 28% male, 3% non-binary/other; mean age 21.5 years, SD 6.0, range 18–51) participated in online testing in exchange for partial research credit in an undergraduate Introductory Psychology course. Eligibility criteria were being eighteen years of age or older, ability to read English and sufficient computer literacy to be able to complete the computer inventories and tasks online. There were no exclusion criteria, other than completion of the questionnaires and both MCQ and GNG tasks (described below). The study was approved by the Institutional Review Board at University of Northern Colorado (protocol # 2202035445) and conformed to the Declaration of Helsinki as well as U.S. Federal Guidelines for the protection of human participants.

An a priori power analysis utilizing G*Power (Faul et al., 2007) indicated that a sample size of 45 participants for each group would provide sufficient power of 95% for a medium effect size. Given expected rates of TBI in the population (estimates ranging about 20–40%), we recruited 116 participants to ensure we would have sufficient numbers in each group. A post-hoc power analysis utilizing G*Power indicated that the final sample sizes for the HI+ (n = 66) and HI- (n = 50) groups provided acceptable power of 84% with a medium effect size.

Procedures

Participants provided informed consent by reading an informed consent statement and agreeing to voluntarily complete the study at the start of the Qualtrics survey. Participants then completed a short demographic questionnaire that included items on self-reported sex/gender, race and ethnicity, and age, as well as a history of anxiety (“Have you ever been diagnosed with, or received treatment for, an anxiety disorder?”), depression (“Have you ever been diagnosed with, or received treatment for, clinical depression?”), and head injury (“Have you ever experienced a head injury or concussion resulting in loss of consciousness or altered state of consciousness (like ‘seeing stars’)?”). All participants responding ‘yes’ to the head injury question were assigned to the HI+ group, while all participants responding ‘no’ to the head injury question were assigned to the HI- group.

A subset of participants also completed additional neurocognitive tasks and questionnaires; these data are not described here.

All participants also completed a Go/No-Go task and an online version of the Monetary Choice Questionnaire (described further below), as well as the UPPS Impulsive Behavior Scale (Whiteside & Lynam, 2001), a 20-item questionnaire that produces subscales on five domains of impulsivity: negative urgency, (lack of) premeditation, (lack of) perseverance, sensation-seeking, and positive urgency. The authors have permission to use the MCQ and UPPS from the copyright holders.

Monetary choice questionnaire

The Monetary Choice Questionnaire (MCQ; Kirby, Petry & Bickel, 1999) is a 27-item questionnaire that queries participants whether they prefer a series of small, immediate rewards or larger, delayed rewards (e.g., “Would you prefer $54 now or $55 in 117 days?”). Based on the participant’s responses, a k-value was computed that describes the slope of a hyperbolic discounting function: i.e., how fast the subjective value of a reward decreases with time (larger values of k indicate steeper discounting, i.e., less willingness to wait). We implemented the 27 questions of the MCQ as questions to be delivered online via the Qualtrics platform. Scoring followed Kaplan et al. (2016) in computing a k-value (geometric mean of k computed for small, medium, and large rewards); because k-values are highly skewed, scores are reported as negative log-transformed (-ln(k)), producing a “willingness-to-wait” measure, where larger positive values mean more willingness to wait for the larger delayed reward (i.e., less cognitive impulsivity). We also report a consistency measure C (indicating how well the discounting curve described by k accurately predicted the participant’s responses), and the proportion of choices for the larger, delayed reward over the smaller, immediate reward.

Go/No-go task

The Go/No-go task followed the methods of Mostofsky et al. (2003). In brief, participants viewed rapidly presented target stimuli (green circles) and were instructed to respond to each by pressing the spacebar as rapidly as possible, but to withhold responding to infrequent foil stimuli (red Xs). Stimuli appeared at the center of the screen, against a black background, and would appear about 1.5″ high on an 18″ monitor. Stimuli remained onscreen for 200 msec, followed by a 1,300 msec inter-trial interval. Keypresses occurring within 50 msec of stimulus onset were discarded as anticipatory responses (or very late responses to the prior trial); otherwise, if at least one response occurred before the onset of the next trial, a Go response was scored, else a No-go response was scored for that trial. The task included 150 trials (including 123 targets intermixed with 27 foils) and was preceded by a short practice phase of 20 trials (16 targets and four foils); practice data were not used for analysis. Trial order was pseudorandom but fixed across participants, with the constraints of no more than two consecutive foils and always at least three consecutive targets. Dependent variables are the percent of misses (failures to respond Go to targets, aka omission errors) and false alarms (failures to inhibit to foils, aka commission errors), as well as mean/SD reaction time of Go responses to targets (correct responses) and foils (false alarms).

Data analyses

Welch’s t-test for independent samples was used to examine effect of self-reported history of HI on the primary dependent measures for each task—willingness-to-wait score on the delay discounting task, percent misses and false alarms on the Go/No-go task, and sub-scores on the UPPS, as well as age and other task variables shown in Table 1; for significant findings, p-values are supplemented with 95% confidence intervals (CI) for the group difference (for all non-significant effects, the 95% CI brackets zero). Where data did not conform to assumptions of normality (Shapiro–Wilk test p < .05), Wilcoxon Rank Sum test was used.

Table 1 Sample characteristics and results.

Summary of demographic and test data for groups with (HI+) and without (HI-) self-reported history of head injury. SD, standard deviation; MCQ, Monetary Choice Questionnaire; Proportion LDR, proportion of choices favoring larger delayed reward; GNG, Go/no-go task; RT, reaction time; UPPS, Impulsive Behavior Scale.

N (and %)	HI- (n = 65)	HI+ (n = 47)	Full sample (n = 112)	
Gender				
- Female	49 (75.4%)	28 (59.6)	77 (68.8%)	
- Male	14 (21.5%)	17 (36.2%)	31 (27.7%)	
- Non-binary/other	2 (3.0%)	2 (4.3%)	4 (3.6%)	
Race				
- White	52 (80.0%)	42 (89.4%)	94 (83.9%)	
- Black or African American	5 (7.7%)	0 (0.0%)	5 (4.5%)	
- Native American or Alaska Native	5 (7.7%)	0 (0.0%)	5 (4.5%)	
- Asian	1 (1.5%)	0 (0.0%)	1 (0.9%)	
- Other/Mixed	2 (3.1%)	5 (10.6%)	7 (6.2%)	
Ethnicity				
- Not Hispanic/Latinx	41 (63.1%)	42 (89.4%)	83 (74.1%)	
- Hispanic/Latinx	24 (36.9%)	5 (10.6%)	29 (25.9%)	
History of anxiety	21 (32.3%)	22 (46.8%)	43 (38.4%)	
History of depression	16 (24.6%)	17 (36.2%)	33 (29.5%)	
Mean (SD)				
Age (years)	21.3 (5.6)	21.6 (6.7)	21.4 (6.1)	
MCQ: Willingness-to-wait	5.0 (1.5)	4.2 (1.1)	4.7 (1.4)	
MCQ: Consistency	1.0 (0.0)	1.0 (0.1)	1.0 (0.0)	
MCQ: Proportion LDR	0.56 (0.2)	0.47 (0.2)	0.52 (0.2)	
GNG: % misses	4.0 (5.3)	4.0 (4.5)	4.0 (4.9)	
GNG: % false alarms	14.5 (11.7)	14.1 (11.9)	14.4 (11.7)	
GNG: RT on correct Go (sec)	0.34 (0.04)	0.35 (0.03)	0.35 (0.03)	
GNG: RT on false alarms (sec)	0.31 (0.04)	0.31 (0.04)	0.31 (0.04)	
GNG: short responses	1.0 (1.9)	0.9 (1.7)	0.9 (1.8)	
UPPS: Negative urgency	9.8 (2.9)	10.5 (2.8)	10.1 (2.9)	
UPPS: Lack of perseverance	7.1 (2.1)	7.3 (2.3)	7.1 (2.2)	
UPPS: Lack of premeditation	7.0 (2.2)	7.8 (2.4)	7.3 (2.3)	
UPPS: Sensation seeking	9.8 (2.9)	11.0 (3.2)	10.3 (3.1)	
UPPS: Positive urgency	8.1 (2.9)	8.8 (3.1)	8.4 (3.0)	

Additional demographic/clinical variables shown in Table 1 (self-reported gender, race, ethnicity, and history of anxiety or depression) were compared between groups using Fisher’s Exact Test for categorical variables.

To examine within-subject correlations among tasks and impulsivity measures, the non-parametric Spearman’s r was used to examine correlations among the primary dependent measures (willingness-to-wait on the delay discounting task, percent misses and false alarms on the Go/No-go task, and sub-scores of the UPPS).

Finally, as a reality check that the GNG task and response times obtained via Qualtrics conformed to expected metrics for this speeded response task, we examined within-subject rates of misses vs. false alarms (expecting higher rates of false alarms than misses) as well as reaction time to correct Go responses vs. false alarms (expecting slightly faster reaction time on false alarm trials, which represent failures to withhold or inhibit responding, than on correct Go responses). In both cases, the within-subject difference scores were non-normally distributed (Shapiro–Wilk test, both p < .05), so the non-parametric paired-samples Wilcoxon test (aka Wilcoxon signed rank test) was used on each set of difference scores, to test whether scores differed from zero.

Significance was defined as p < .05, except for the within-subject correlations shown in Table 2 (28 tests), where Bonferroni correction was applied (requiring p<.0017 for significance).

Table 2 Correlations between measures of impulsivity (Spearman’s r).

Correlations (Spearman’s r) between measures of impulsivity; r reported as 0 if −0.001<r <0.001. Other abbreviations as in Table 1.

	GNG:
% Misses	GNG:
% False alarms	UPPS: Negative urgency	UPPS:
Lack of perseverance	UPPS:
Lack of premeditation	UPPS: Sensation seeking	UPPS: Positive urgency	
MCQ: willingness-to-wait	0.16	−0.02	−0.07	0.10	−0.07	0.06	0.03	
GNG: % misses		0.13	0.04	−0.05	0.17	0.07	0.06	
GNG: % false alarms			0	0	0.18	0.02	0	
UPPS: negative urgency				0.18	0.09	−0.06	0.52**	
UPPS: lack of perseverance					0.42**	−0.08	0.08	
UPPS: lack of premeditation						0.06	0.22*	
UPPS: sensation seeking							0.28**	
Notes.

* p < .05.

** p < .005.

Three participants (two HI+ and one HI-) made more than 50% misses on GNG (two made > 98% misses), suggesting they may not have been attending to the task; an additional participant (in the HI+ group) had consistency C < 60% on the delay discounting task, meaning that the hyperbolic curve did not provide a good fit to the participant’s responses (Kaplan et al., 2016), again suggesting possible lack of attention. Data from these participants were excluded from the analysis; however, overall study results were similar when these participants’ data were included.

The full dataset (n = 116), and R Script generating the results and figures presented in this article, are available at https://osf.io/52f47/.

Results

Table 1 shows the demographic, questionnaire, and task results, for subjects self-reporting history of head injury (HI+, n = 47) vs. no history of head injury (HI-, n = 65). The HI- and HI+ groups did not differ in age (Wilcoxon Rank Sum test, W = 1,431, p = .56), gender distribution (Fisher’s Exact Test, p = 0.16) or in rates of anxiety or depression (Fisher’s Exact Test, both p > 0.1). However, the groups did differ significantly in ethnicity, with more Hispanic/Latinx participants in the HI- group (Fisher’s Exact Test, p = .002), and in distribution of race (Fisher’s Exact Test, p = .013), with relatively more White participants in the HI+ group than the HI- group.

On the delay discounting task (MCQ), participants with HI+ had significantly reduced willingness-to-wait, as shown in Fig. 1A (Wilcoxon Rank Sum Test, W = 2,020, p = .004 [0.31, 1.24]). The groups did not differ in consistency (Wilcoxon Rank Sum Test, W = 1,410, p = .47 [−0.04, <0.01), although the HI+ group had a smaller proportion of choices for the larger, delayed reward (Wilcoxon Rank Sum Test, W =1,925, p = .017 [<.01, 0.11]), consistent with reduced willingness-to-wait in the HI+ group.

Figure 1 Results on tests of cognitive and motor impulsivity in the HI- and HI+ groups.

(A) On delayed discounting, the HI+ group showed significantly reduced willingness-to-wait (-ln(k)), compared to the HI- group. (B, C) On Go/No-go, there were no significant differences between groups in either misses (failures to Go to target stimuli) or false alarms (failures to inhibit response to foils).

On Go/No-go, as expected, the rate of false alarms was higher than the rate of misses within-subjects (paired-samples Wilcoxon test, V = 5, 718, p < .001 [7.37,11.69]), and the mean reaction time was faster for false alarms than for correct Go responses (paired-samples Wilcoxon test, V = 5, 568, p < .001 [0.035, 0.045]), consistent with false alarms representing failures to inhibit (or stop) a prepotent response.

However, there were no differences between HI+ and HI- groups in either miss rate (Fig. 1B; Wilcoxon Rank Sum test, W = 1,406, p = .47) or false alarm rate (Fig. 1C; Wilcoxon Rank Sum test, W = 1,605, p = .65), and no group differences in mean reaction time on either correct Go responses (Welch’s t-test, t(109) = 1.26, p = .209) or false alarms (Wilcoxon Rank Sum test, W = 1,293, p = .49). Overall, the rate of anticipatory responses (occurring < 50 msec after stimulus onset) was low, but the number of trials excluded due to short RT did not differ between groups (Wilcoxon Rank Sum test, W = 1,554, p = .86).

On the UPPS, Sensation Seeking was higher in the HI+ group than the HI- group (Wilcoxon Rank Sum test, W = 1,178, p =.038 [−3.00, −5.65e−5]); no other scores differed between groups (Wilcoxon Rank Sum tests, all p > 0.05).

Table 2 shows within-subjects correlations between the key behavioral measures. Within the UPPS sub-scores, expected correlations were observed between sub-scores for Negative Urgency and Positive Urgency (p < .001) and between Lack of Perseverance and Lack of Premeditation (p < .001), with the relationship between Sensation Seeking and Positive Urgency approaching corrected significance (p =.003). However, no relationships between delay discounting and GNG, or between either of these tasks and UPPS subscores, approached uncorrected significance (all p > .05).

Discussion

The major findings of the current study were that healthy undergraduates self-reporting a head injury exhibited increased cognitive impulsivity, as measured by less of a willingness to wait for a larger reward in a delay discounting task, but no significant differences in motor impulsivity in a Go/No-go task. These results in a non-clinical, young, online sample, self-reporting a history of HI, replicate those in a clinical sample carefully assessed for mTBI vs. no history of TBI (Interian et al., 2024), where history of mTBI was also associated with increased cognitive impulsivity but not increased motor impulsivity on a Go/No-go task.

Our current findings also fit with some of the prior findings of some form of increased impulsivity following TBI. Our finding in which HI+ individuals exhibited increased cognitive impulsivity in the delay discounting task fits with prior reports by McHugh & Wood (2008) and Richards et al. (1999) in which adults with a history of TBI displayed less willingness to wait for delayed rewards, compared to participants with no history of TBI. However, our finding of no increased motor impulsivity on the Go/No-go task does not match prior studies that have reported that history of TBI is associated with poorer performance on Go/No go tasks (e.g., Gagnon et al., 2006; Dimoska-Di Marco et al., 2011). Similarly, although there were expected correlations between self-reported impulsivity on the UPPS sub-scores, most strongly between Negative Urgency and Positive Urgency, as has been previously reported with the UPPS (Billieux et al., 2012; Cándido et al., 2012; Claréus et al., 2017; Cyders et al., 2014; Cyders & Smith, 2007; Pedersen et al., 2016; Riley & Smith, 2017), there were no strong relationships between UPPS subscales and either willingness-to-wait on the delay discounting task, or impulsive responding (failures to inhibit) on the Go/No-go task.

Taken together, these findings suggest either that cognitive and motor impulsivity are different domains within the broader construct of impulsivity, or else that participants’ performance on tasks that assess impulsivity is different from their self-report. They also suggest that association of cognitive impulsivity, specifically willingness to wait for delayed reward, with head injury is observed in multiple populations, spans multiple types of HI severity, including mild TBI, is not a transient effect observed only in the acute aftermath of injury, and persists even in putatively healthy, high-functioning individuals.

The greatest limitation of the current study is that it did not discriminate between severity of head injury (e.g., concussion or moderate TBI vs. moderate/severe TBI), did not consider recency or number of injuries, and scored HI history based on self-report of one question rather than clinical assessment and/or medical records. Hux, Brown & Lambert (2017) separated individuals with HI into three categories with participants in a high symptomatology, moderate, and a negligible symptomatology category. The symptoms that best separated the high and moderate classes were problems with memory including slow thinking, difficulty learning new information, and problems with attention. Those individuals within the high symptomatology category experienced greater academic challenges than those with moderate and negligible symptoms. Future work could collect more details on post-HI symptoms in undergraduates which would allow for analysis of the effects of these specific symptoms on cognitive impulsivity along with possible academic issues that these college students may be experiencing. However, the fact that noisy self-report data revealed the same relationship with cognitive impulsivity as in the prior clinical sample speaks to the potential strength of the effect of head injury/TBI on cognitive impulsivity.

The current study was also conducted wholly online, which means that some participants may have been distracted or inattentive. For example, three participants likely were not attending during the Go/No-Go task, as evidenced by extremely high miss rates—suggesting they were not attempting to make Go responses but rather letting the task “time out”. Additionally, the delivery of the GNG task over the Internet meant that there was likely loss of precision in the reaction times recorded, as well as variability related to individuals’ hardware and connection speeds. Still, sufficient precision was able to detect the expected within-subjects difference between correct and incorrect Go responses (with errors being on average about 200 msec faster than correct responses), indicating that the loss of precision was small relative to the overall effect. Still, these considerations mean that the finding of no group difference on GNG may reflect failure to detect rather than true absence of difference.

Limited variability in gender and age of the current sample of undergraduates may limit generalizability of the current findings to other populations. Future work should utilize a more balanced ratio of males to females that may provide increased power in identifying gender differences in head injury and impulsivity, such as the finding that male patients tend to exhibit more impulsive behaviors than female patients following TBI (Willer et al., 1991). Additionally, the current sample considered young (college-age) adults; the prefrontal cortex in particular may not fully mature until the third decade of life (for review see Kolk & Rakic, 2022) and is known to play a role in impulsivity (Spinella, 2004). Future work could assess the relationship of HI/TBI to cognitive and motor impulsivity across various points in the lifespan to further explore developmental issues in response to head injuries.

Despite the above limitations, the finding of reduced willingness-to-wait in a non-clinical sample of undergraduates with self-reported history of HI, when considered together with highly similar findings in a clinical sample with carefully documented mild TBI history, provides converging evidence for the relationship between cognitive impulsivity and head injury.

Conclusions

The current study sought to test the hypothesis that a non-clinical sample of undergraduates self-reporting a head injury would show alterations in cognitive impulsivity but not motor impulsivity, thus partially replicating and extending prior findings from Veterans clinically diagnosed with mTBI. Indeed, putatively healthy individuals with a self-reported history of head injury exhibited increased cognitive impulsivity, in that they were less likely to choose larger, delayed rewards over smaller, immediate rewards on a delay discounting task, but did not display increased motor impulsivity (i.e., failure to inhibit responses on a Go/no-go task). Together, the results indicate the strength of the relationship between head injury/TBI and changes in cognitive, but not motor, impulsivity. Given these findings, this work should be continued with additional attention given to the details of the head injury such as the type of head injury, age at the time of the injury, whether there was a loss of consciousness, whether it was a single or repeated head injuries, in order to further explore how various forms of head injury including TBI can produce in changes in cognitive impulsivity that may underlie various psychopathologies.

Supplemental Information

Supplemental Information 1 Demographic, inventory, and task data

Supplemental Information 2 STROBE checklist

Additional Information and Declarations

Competing Interests

Author Contributions

Human Ethics

Data Availability

C.E. Myers is an Academic Editor for PeerJ.

M. Todd Allen conceived and designed the experiments, performed the experiments, analyzed the data, authored or reviewed drafts of the article, and approved the final draft.

Alejandro Interian conceived and designed the experiments, authored or reviewed drafts of the article, and approved the final draft.

Vibha Reddy analyzed the data, authored or reviewed drafts of the article, and approved the final draft.

Kailyn Rodriguez conceived and designed the experiments, authored or reviewed drafts of the article, and approved the final draft.

Catherine E. Myers conceived and designed the experiments, analyzed the data, prepared figures and/or tables, authored or reviewed drafts of the article, and approved the final draft.

The following information was supplied relating to ethical approvals (i.e., approving body and any reference numbers):

University of Northern Colorado Institutional Review Board (protocol # 2202035445)

The following information was supplied regarding data availability:

The data for the demographics, inventory responding, and impulsivity tasks are available in the Supplementary File.

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
