# Peer review of "Self-reported history of head injury is associated with cognitive impulsivity on a delay discounting task"

_PeerJ, doi:10.7717/peerj.19057_

## Round 0.1 · original submission · Major Revisions

As you can see, the reviewers have provided thorough sets of comments, amounting to a need to perform major revisions. Please attend to their suggestions.

·

Basic reporting

Abstract: In lines 33-34, I suggest adding "cognitive problems" alongside mental and physical issues, as the primary focus of the study is to investigate cognitive challenges following TBI (e.g., cognitive impulsivity).

Background: The introduction section includes a significant amount of information that may not be directly relevant to the study's focus.
For example, lines 66 to 85 provide extensive information on the prevalence of TBI across different populations, including variations by sex, ethnicity, etc. However, these differences do not seem directly relevant to the purpose of the study.
Similarly, lines 94 to 99, which discuss the somatosensory effects of TBI, are not relevant to the focus of this study.
I recommend summarizing this content and focusing on the most pertinent information for the study. Doing so would enhance the clarity of the study’s purpose and improve the flow of the text.

Experimental design

Hypothesis: I suggest renaming this section "Aim and Hypothesis." Additionally, I recommend revising and summarizing the content to avoid repetition.

Procedure:
- Did the authors include some exclusion criteria for participant selection?

- It would be more coherent to include the sample size calculation in the "Participants" section. Lines 216-220: “An a-priori power analysis utilizing G*Power (Faul et al., 2007) indicated that a sample size of 45 participants for each group would provide sufficient power of 95% for a medium effect size. In addition, a post-hoc power analysis utilizing G*Power indicated that the final sample sizes for the HI+ (n= 66) and HI- (n=50) groups provided acceptable power of 84% with a medium effect size.”

- The authors state: “Three participants (two HI+ and one HI-) made more than 50% misses on GNG (two made >98% misses), suggesting they may not have been attending to the task; an additional participant (in the HI+ group) had consistency C<60% on the delay discounting task, meaning that the hyperbolic curve did not provide a good fit to the participant.s responses(Kaplan et al., 2016), again suggesting possible lack of attention.”. They mentioned that the results remained unchanged whether these participants were included or excluded from the analyses, so they chose to include them. However, to maintain the reliability of the results, these participants should be excluded, as they did not perform the task properly.

- The analyses of sociodemographical variables and rates of reported diagnosis of anxiety or depression are not described in the “Data analyses” section.

- The regression analysis is not described in the “Data analyses” section.

- In the “data analyses” section the authors say “Additionally, the non-parametric Spearman’s r was used to examine correlations among these measures.” However, the variables included in these correlations are not specified, and the purpose of conducting these correlation analyses is unclear. Later, in the "Discussion" section, it becomes clear why correlations between MCQ and GNG false alarms were performed. However, the rationale for the correlations between the UPPS sub-scores remains unclear.

Validity of the findings

Results:
- Please refer to Table 2 in the text when reporting the results of the correlation analyses.

- The results of the regression are incomplete. Was the model significant? Please provide R2 and beta values.

Discussion:
- Lines 277-284. This issue is already addressed in detail in the limitations section, so I recommend removing it from this part of the discussion.

- The authors state (lines 293-296) “In addition, the finding that HI+ individuals had higher scores for sensation seeking on the UPPS fits with McAllister’s (2008) definition of impulsivity as well as the theory that impulsivity may occur about due to an exaggerated sensitivity to reward (Gray, 1970, 1981; Madden & Johnson, 2010).” The authors should refrain from making that assertion based on their results. They are not comparing sensation-seeking behaviors between more impulsive and less impulsive participants, but rather between HI+ and HI-. Therefore, they cannot conclude that more impulsive participants exhibit greater sensation-seeking behaviors, as the previous sentence seems to suggest. In fact, the correlation analysis indicates that such an association is not present. I recommend revisiting this sentence for clarity.

- It is not entirely clear why the authors wish to compare this study with the one conducted on veteran participants concerning gender and age variables. I understand that the present study's hypothesis is based on that study and that the results somewhat replicate those of Interian et al. (submitted). However, the findings of the current study are robust on their own, and a strict comparison with Interian et al. may not be necessary, especially considering the main difference lies in the inclusion of a clinical population in their study versus a non-clinical population in the present study. Instead of emphasizing the inability to compare with Interian et al.'s study, it would be more appropriate to address the limitation of a lack of variability in age and gender, as this may affect the generalizability of the results of the current study.

Additional comments

Overall, this article provides valuable insights into the field of cognitive impairments in healthy individuals with a history of head injury. The selection of variables is appropriate and the methodology is thoroughly considered (open data, corrections for multiple comparisons, sample size calculation, etc.). The results are noteworthy, indicating an increase in cognitive impulsivity among these participants, which may significantly affect their daily lives.

With some revisions to enhance clarity and depth, this study has the potential to make a substantial impact in the field.

Reviewer 2 ·

Basic reporting

The introduction could benefit from reorganization and tightening up. Over a page of introduction provide statistical factoids about TBI and head injuries that don’t appear to have any bearing the study design/rationale. I recognize some orientation to the topic issue is important, but could be shortened and more on-point. There is a lack of narrative to guide the reader to the purpose of this study, often there are paragraphs without transitions and just blocks of summaries of studies.

The introduction should, briefly, discuss the different measures of impulsivity (questionnaires, MCQ, GNG) and how they typically correlate in past studies (i.e., they tend to be distinct). Since this work is related to the study with Veterans, there should be more introduction of that study and how its results shape expectations of this study. I recognize that study was “submitted” and that is a complication, but since this study seems anchored to that study (“Here, we attempted to extend the results of Interian…” line 145), I feel this is important.

While the HI+ and HI- groupings appear to be fairly self-evident, it would be helpful for the authors formally state how the groups were determined (e.g., All ‘yes’ responses to the “Have you ever experienced a head injury or concussion resulting in loss of consciousness or altered state of consciousness?” were assigned to the HI+ group and all ‘no’ responses were assigned to the HI- group.).

Like the introduction, the discussion would benefit from a more narrative flow with transitions and fluid organization.

Figure 1 labels were difficult to read, the font needs to be increased relative to the bar size.

The format of the statistical reporting is unfamiliar to me, often the CI is enclosed in brackets and the stat test is italicized.

I did catch a few random typos, so a copy-editing review by the authors is suggested.

Experimental design

No comment.

Validity of the findings

For the Go/No-Go task, were responses only during the 50 msec window excluded and how many responses did this affect? I assume it was very rare, but if it was common then it would be data and should be included in some way (i.e., are HI+ more likely to make these premature/late responses?).

Authors discuss accommodations for assumptions of equal variance, but it is unclear about assumptions of normality. The percentage of misses and failure to inhibit data are constrained between 0 and 100, were the data normally distributed or skewed away from the floor? If so, then a Poisson regression would better model group differences in mean number of misses and false alarms. It is necessary to know about this before accepting the claim that GNG data don’t differ (Figure 1B shows a trend for HI+ showing more errors, a different model might capture that).

Additional comments

Line 293-295: I am not following the reasoning linking sensation seeking to sensitivity to reward. As written, it sounds like speculation. If so, it should be removed.

Lines 316-326: These limitations are endemic to all studies of this sort, unless there is reason to believe they are uniquely affecting the data in the present study, then it doesn’t need to be discussed here.

Lines 327-331: This is an interesting point, but I would suggest framing it as a ‘future directions we will look at ___ as a possible intervening/moderating variable’ (perhaps directly linked to the discussion in the following paragraph). Otherwise, it is just telling the reader what they should already know, that association does not imply causation.

Lines 343-351: This is precisely the sort of comparison of present data and other studies I’d like to see in the discussion. But I would like to see a more direct comparison of the results such as the MCQ, GNG, and other collected data to show the reader what the present study is adding to the literature.

Lines 352-364: This section could be shortened since it doesn’t really discuss the current study much. A shortened version could be retained as a brief mention of future directions (but should be clear how such a future direction builds upon the present study).

Lines 365-373: I am confused by this section. Are comparisons between undergraduate and veteran problematic because of education? There are many population differences (as noted by author), why is this one a problem? Also, the relationship between intelligence and discounting is mixed and so the citation of Monterosso shouldn’t be made without reference of other publications concluding differently (e.g., Shamosh et al., 2008).

Lines 374-384: There are other studies looking at that temporal discounting and SES. And, I am not sure how this relates to the present study with head injuries and temporal discounting? This section should be linked to the current study or excluded.

---

## Round 0.2 · accepted · Accept

Thank you for your response to the reviewer's comments. You have made amendments that have satisfied the reviewer's queries and areas for improvement. I suggest that a final review for typos is completed as part of the proofing process to ensure consistency in terminology and the highest degrees of correctness.

·

Basic reporting

The authors have done a great job reviewing the manuscript and incorporating the suggested modifications. I have no further comments and the manuscript can be accepted.

Experimental design

No comment

Validity of the findings

No comment

Additional comments

No comment

Reviewer 2 ·

Basic reporting

Overall the basic reporting is good. However, in Figure 1 it would be helpful if the bars are defined as HI- and HI+ and not 0 and 1 on the x-axis. Also, a final review might be helpful to catch a few stray typos, I caught a "Hi" rather than "HI" in one instance.

Experimental design

No Comment

Validity of the findings

No Comment

Additional comments

Overall, this revision is much improved and ready for publication.